# Enhanced Silk Fibroin/Sericin Composite Film: Preparation, Mechanical Properties and Mineralization Activity

**DOI:** 10.3390/polym14122466

**Published:** 2022-06-17

**Authors:** Meng Li, Wei Tian, Yao Zhang, Hui Song, Yangxiao Yu, Xiangshang Chen, Nan Yong, Xiuzhen Li, Yin Yin, Qingmin Fan, Jiannan Wang

**Affiliations:** 1College of Textile and Clothing Engineering, Soochow University, Suzhou 215123, China; 20204215052@stu.suda.edu.cn (M.L.); 20175215025@stu.suda.edu.cn (W.T.); 20215215005@stu.suda.edu.cn (Y.Z.); 20215215001@stu.suda.edu.cn (H.S.); 20214215008@stu.suda.edu.cn (Y.Y.); cxs117383@163.com (X.C.); yongnan202203@163.com (N.Y.); lxz20020111625@163.com (X.L.); 2Department of Medicine, Soochow University, Suzhou 215123, China; yinyin@suda.edu.cn; 3Department of Ultrasonography, First Affiliated Hospital, Soochow University, Suzhou 215006, China; fanqingmin@suda.edu.cn

**Keywords:** silk fibroin, sericin, high molecular weight, composite film, mineralization

## Abstract

The periosteum plays an important role in bone formation and reconstruction. One of the reasons for the high failure rate of bone transplantation is the absence of the periosteum. Silk fibroin (SF) and silk sericin (SS) have excellent biocompatibility and physicochemical properties, which have amazing application prospects in bone tissue engineering, but lacked mechanical properties. We developed a series of SF/SS composite films with improved mechanical properties using boiling water degumming, which caused little damage to SF molecular chains to retain larger molecules. The Fourier transform infrared spectroscopy and X-ray diffraction results showed that there were more β-sheets in SF/SS films than in Na_2_CO_3_ degummed SF film, resulting in significantly improved breaking strength and toughness of the composite films, which were increased by approximately 1.3 and 1.7 times, respectively. The mineralization results showed that the hydroxyapatite (HAp) deposition rate on SF/SS composite films was faster than that on SF film. The SF/SS composite films effectively regulated the nucleation, growth and aggregation of HAp-like minerals, and the presence of SS accelerated the early mineralization of SF-based materials. These composite films may be promising biomaterials in the repair and regeneration of periosteum.

## 1. Introduction

Bone defects are commonly caused by fractures, osteoporosis, bone infections, tumors, or congenital defects [1]. In regenerative medicine, clinical bone transplantation faces great challenges, and one of the main reasons for the high failure rate is the loss of periosteum [2].

Periosteum is the highly vascularized tissue that covers the outer surface of almost all cortical bone. A large number of studies have shown that periosteum plays a crucial role in the healing and remodeling of bone defects as it is a store for progenitor cells, a source of local growth factors and a scaffold for recruiting cells and growth factors. Removal of periosteum from bone autografts results in a marked reduction in new bone and cartilage formation on the graft [3,4,5,6]. However, due to traumatic, congenital and surgical reasons, it is always difficult to retain intact periosteum; therefore, artificial materials such as artificial periosteum are needed to replace its effects. 

Currently, scaffolds fabricated using various materials have been explored as substitutes for periosteum, including extracellular matrix (ECM) scaffolds, cell sheets, natural polymer materials and synthetic polymer materials. Tissue-derived ECM scaffolds, mainly include small intestinal submucosa (SIS) and acellular dermis, which comprise retained proteins, proteoglycans and growth factors [7]. SIS is considered an attractive candidate for periosteum tissue engineering construction due to its elastic, immunocompatible, and biodegradable properties [8,9]. In recent years, acellular dermis, such as amniotic membrane, porcine membrane and human acellular materials, have been used to replace periosteum and to treat bone defects [10,11,12]. It is a simple and convenient method to simulate periosteum tissue by covering the bone defect with a multipotent stem cells sheet. Studies have shown that ECM derived from marrow mesenchymal stem cells (MSCs) could promote proliferation, osteogenesis differentiation and chondrogenic differentiation of reseeded MSCs [13,14].

The most widely studied natural polymer materials are polysaccharide polymers and polypeptide polymers, such as chitosan and collagen membranes which are widely used as bone-healing materials [15,16,17]. However, natural biopolymers lack mechanical properties and do not meet the requirements for temporary mechanical substitution of periosteum tissue. The most commonly used synthetic polymers for periosteum development include polylactic(L-lactic-co-glycolic acid), polyurethanes, poly(ethylene glycol), polycaprolactone and poly(L-lactic acid) [18,19,20,21,22]. However, these synthetic materials also have some disadvantages: they can induce host tissue response and foreign body reactions during degradation (by non-enzymatic hydrolysis), and moderate cytotoxic reactions may reduce cellular adhesion [23].

Among the various bioengineered materials, SF has attracted more and more attention due to its good biocompatibility, adjustable biodegradability and low immunogenicity [24,25]. To assess its bone regeneration capacity, SF has been tested as a porous scaffold, electrospun material and a hydrogel in various animal models but mainly in rats and rabbits with calvarial or femoral defects [26]. The results have shown that SF-based biomaterials could effectively promote bone regeneration. However, studies have shown that the regenerated SF materials also lacked mechanical properties [27,28]. In addition, studies have shown that the complete removal of SS reduced the strength of silk fibers [29] and the degumming process conditions could substantially change the physicochemical and mechanical properties of SF.

Silk sericin (SS) which is found in the outer layer of silk is also a natural macromolecular protein, which is mostly removed as waste in the application of silk products, resulting in environmental pollution [30]. SS contains a large number of hydrophilic amino acids with -OH and -COOH groups, such as aspartic acid (Asp), serine (Ser) and glutamic acid (Glu). The presence of these functional groups in SS can provide sites for the deposition of calcium phosphate and biomineralization by binding Ca^2+^ through electrostatic and coordination interactions [31]. SS as a bone tissue engineering scaffold comes in two main forms, a hydrogel and porous material, and the hydrogel is commonly used in cartilage tissue engineering [32]. In addition, SS can also be prepared as membrane material to support the adhesion and proliferation of osteocytes [33]. Therefore, an appropriate degumming method to retain part of SS might be used to make materials suitable for bone tissue.

Based on the research status and existing problems of silk protein in bone tissue engineering materials, this research aimed to develop novel SF/SS composite films with improved mechanical properties and mineralization activity. We used an environmentally friendly and mild boiling water degumming method to obtain SF with a concentrated distribution of high molecular weight, which endowed excellent mechanical properties in the regenerated SF-based films. The SF/SS composite films containing different SS contents were obtained by controlling the degumming time. The chemical composition and structure, crystallization property and mineralization ability were then explored. This study provided a new simple method to obtain a biomimetic artificial periosteum with improved mechanical properties, and had positive significance for environmental protection.

## 2. Materials and Methods

### 2.1. Materials

*Bombyx mori* raw silk (20/22D, 6A) was purchased from HaiAn Tian Xin Silk Industry (Nantong, China), Cellulose membrane (14,000 Da, MD44) was purchased from Union Carbide (Danbury, CT, USA). The chemicals such as Sodium carbonate (Na_2_CO_3_, ≥99%), Calcium chloride (CaCl_2_, ≥98%), Dibasic sodium phosphate (Na_2_HPO_4_, ≥98%), and Coomassie Brilliant Blue (≥55%) Tris-(hydroxymethyl) aminomethane were purchased from Sinopharm Chemical Reagent Co., Ltd. (Shanghai, China). Lithium bromide (LiBr, ≥99%) was purchased from Finecollection Institute of Chemical Industry (Hefei, China). Sodium dodecyl sulfate (SDS, ≥99%), Ammonium persulphate (≥99%), Acrylamide (≥99%), N,N,N′,N′-Tetramethylethylenediamine (≥99%), Tris Base were purchased from Sigma Aldrich (St. Louis, MO, USA).

### 2.2. Degumming and Dissolving of Silk

*Bombyx mori* raw silk was, respectively, treated in boiling distilled water at the bath ratio of 1:50 (*w*/*v*) for 1 h, 2 h, 3 h and 5 h (Table 1), and the boiling distilled water was replaced every hour. Na_2_CO_3_ (wt 0.05%) degumming was used as a control group [34]. After drying, the degummed silk was dissolved in 9.3 mol/L LiBr at 65 °C for 1 h. Then the dissolved silk solution was dialyzed against distilled water at 4 °C for 72 h using a cellulose membrane to remove LiBr (Appendix A). The final silk protein aqueous solution was concentrated to 40 mg/mL.

### 2.3. Morphological Characterization of Degummed Silk

The degummed silk was cut with a blade, and the specimens were subjected to natural drying. After sputtering with gold for 90 s, the morphological structure of the samples was observed using a TM3030 scanning electron microscope (Hitachi, Japan).

### 2.4. Preparation of SF/SS Composite Films

All films were prepared by pouring the same amount (9 mL) of silk protein solution (40 mg/mL) into the same area of a polyethylene dish (diameter 50 mm) to debubble, and then dried at 40 °C by revolving slowly for 6 h. The SF/SS composite films and SF film were immersed in 80% (*v*/*v*) ethanol for 2 h and washed with distilled water (Appendix A). The films without ethanol treatment were used as controls.

### 2.5. Amino Acid Composition Analysis

Amino acid analysis was performed by hydrolyzing proteins in 6 mol/L HCl according to previous studies [35]. Protein samples were dissolved in HCl at 110 °C for 24 h and dried at 65 °C. After redissolving in 0.02 mol/L HCl, the amino acid composition was determined using a Hitachi L-8800 Amino Acid Analyzer (Hitachi, Japan).

### 2.6. SDS-PAGE

The molecular weight and distribution of the silk protein solution were assessed using SDS polyacrylamide gel electrophoresis (SDS-PAGE, Bio-Rad, Hercules, CA, USA). The concentration of silk protein solution in each group was adjusted to 10 mg/mL, and 15 μL of the samples was subjected to 10% polyacrylamide gel [36]. The gel was stained with Coomassie Brilliant Blue for 1 h and then decolorized until the background color disappeared, leading to the appearance of clear protein bands.

### 2.7. Viscosity Measurement

The concentration of the silk protein solution in each group was adjusted to 18 mg/mL, and the viscosity was measured using an AR2000 rheometer (TA, New Castle, DE, USA). The parameters related to viscosity measurement were selected as follows: Flow mode, 40 mm /1° cone plate, 25 °C, and a sample volume of 200 μL.

### 2.8. Structure Characterization

Chemical structure and molecular conformation of the silk protein films were analyzed using a Nicolet Avatar-IR360 Fourier transform infrared spectroscopy (FTIR, Nicolet, Madison, WI, USA). Thirty-two scans were recorded with a resolution of 4 cm^−1^ and a scanning range of 500–4000 cm^−1^. In addition, we semi-quantitatively analyzed the content of the molecular conformation by performing deconvolution of the amide I band from the FTIR spectra. The crystal structure of all samples was determined by a X‘Pert-Pro MRD X-ray diffractometer (XRD, Philips, Amsterdam, The Netherlands) with a CuKα radiation source in the 2θ of 5–50°, and a scanning speed of 2°/min.

### 2.9. Thermal Property Assay

All samples were in powder form. After filtering through 150 mesh sieves, the thermal properties of the powder samples weighing 3−5 mg were analyzed using a diamond thermogravimetric analyzer (TG-DTA, PerkinElmer, Waltham, MA, USA). The parameters related to thermoanalysis were as follows: temperature range of 30–600 °C, heating rate of 10 °C /min, and a nitrogen atmosphere.

### 2.10. Mechanical Properties Measurement

All films were cut into rectangles of 10 mm × 50 mm and the thickness was measured by an electronic spiral micrometer. After immersing in deionized water for 1 h, the tensile properties of the films were measured in the wet state using an Instron 3365 universal testing machine (Instron, Boston, MA, USA). Parameter setting: clamp distance 20 mm, stretching rate 20 mm/min. For each film, 10 independent samples were tested and the average value was taken. Young’s modulus was obtained by calculating the slope of the initial straight line portion of the stress-strain curve.

### 2.11. Mineralization of SF/SS Composite Films

Mineralization of all films was performed using the alternate soaking method [37]. Briefly, the films were first dipped in 0.5 mol/L CaCl_2_ solution buffered with tris-(hydroxymethyl) aminomethane (pH = 7.45), and then immersed in 0.3 mol/L Na_2_HPO_4_ solution (pH = 8.50) at 37 °C for 30 min, respectively. Afterwards, the premineralized films were placed in simulated body fluid (SBF) at 37 °C which was replaced once a day. The samples were removed at 1 day, 3 days, 5 days and 7 days, washed with distilled water and then dried at room temperature.

### 2.12. Surface Element Assay

The mineralized samples were affixed to the sample stage and sprayed with gold for 90 s, and then scanned using the TM3030 scanning electron microscope (Japan) equipped with a SwiftED3000 X-ray energy spectrometer (EDX, Oxford, UK), in order to obtain the distribution and content of surface elements.

### 2.13. Statistical Analysis

Statistical analysis of values is presented as mean ± SD. Comparisons of means were analyzed by one-way analysis of variance (ANOVA) using SPSS 17.0 statistical software (IBM, Armonk, NY, USA). Statistical significance was set at *p* < 0.05.

## 3. Results and Discussion

### 3.1. Microstructure of Degummed Silk

Silk derived from silkworms contains two major components: SF and SS. Figure 1 shows the microstructure of silk following different degrees of degumming. The surface of raw silk covers a complete layer of SS that binds two fibroin fibers tightly together. Non-degummed silk fibers were closely aligned. After degumming in boiling water for 1 h, the cross-section of fibers was irregular, which was attributed to partial SS shedding as well as discontinuous distribution of the remaining SS. With increased degumming time in boiling water, clear gaps appeared between SF fibers. Following degumming for 5 h, most SS was removed, and the fiber surface was smooth. The surface and cross-section of Na_2_CO_3_ degummed silk fibers were relatively smooth, but not only all fibers were reduced in diameter but also there were some longitudinal grooves on the surface of fibers, indicating that degumming by Na_2_CO_3_ could destroy the microstructure and even the macromolecular chains of SF fibers, which is consistent with previous reports [38].

### 3.2. Amino Acid Composition of SF/SS Composites

Both SF and SS are composed of the same 20 types of amino acids, but their individual amino acid contents differed significantly. SF contains mostly glycine (Gly), alanine (Ala), Ser and tyrosine (Tyr) [39], accounting for more than 80% of the total. In contrast, SS contains more hydrophilic amino acids, including Ser, Asp, threonine (Thr), Glu and so on [40]. We tested the amino acid composition of SF/SS composites prepared using different degumming degrees, which are shown in Table 2. The contents of Gly and Ala in SF/SS composite solutions gradually decreased with increased of SS, because these two amino acids contained in SS (18.59%) were far less than those in SF (74.72%). The contents of Asp, Ser, Thr and Glu in SF/SS composite solutions gradually increased with decreased degumming time, SF/SS_1_ > SF/SS_2_ > SF/SS_3_ > SF/SS_4_. The contents of these four amino acids in SS (64.93%) were significantly higher than those in SF (14.52%). These four amino acid residues carry carboxyl and hydroxyl groups, and provide the structural basis for inducing the deposition of hydroxyapatite (HAp), as these functional groups can bind Ca^2+^ via electrostatic interactions or coordination interaction [34,41]. In addition, tryptophan was completely destroyed by HCl hydrolysis, so no tryptophan was detected in all samples.

### 3.3. Molecular Weight Distribution of SF/SS Composite Solutions

There are three types of peptide chains in the SF macromolecule, light chain, heavy chain and P25 glycoprotein, which are 25 kDa, 325 kDa and 26 kDa, respectively. SS, composed of 18 amino acids, is a water-soluble globular protein with a molecular weight ranging from 10 to 400 kDa. The molecular weight varies depending on the extraction method employed [42]. Silk fiber dissolution is the process in which the force (mainly hydrogen bonding) between the macromolecular chains of silk protein are broken by solvent molecules, and during this process, the amide bonds within the molecular chains of silk protein also undergo different degrees of hydrolysis, producing proteins or polypeptides with different degrees of polymerization. SDS-PAGE was carried out to determine the molecular weight and its distribution in SF/SS solution. Figure 2 shows that the molecular weight of proteins prepared by boiling water degumming was mainly concentrated above 200 kDa. With increased degumming time, a small amount of diffuse distribution appeared in the samples of the SF/SS_4_ group. In the SF/SS_1_ and SF/SS_2_ groups, there were two clear bands, which represented the P25 glycoprotein and SF light chain. As the molecular weight distribution of SF/SS_1_ and SF/SS_2_ were similar, we selected SF/SS_1_ for subsequent research. In the SF/SS_3_ and SF/SS_4_ groups, the P25 glycoprotein was decomposed, and the bands disappeared. The SS molecular weight was high, but close to the SF fiber layer, the SS with smaller molecular weight was distributed (data not shown). However, the SF molecular weight degummed by Na_2_CO_3_ was almost uniformly distributed continuously from above 200 kDa to below 15 kDa. Compared with Na_2_CO_3_ degumming, high molecular weight SF or SF/SS materials could be obtained by boiling water degumming without serious damage to the SF macromolecular chains.

### 3.4. Viscosity of SF/SS Composite Solutions

The rheological behavior of the solution can also reflect the molecular weight and its distribution [43]. As shown in Figure 3, the shear-viscosity curves of all groups showed a similar pattern of variation. The shear viscosity increased sharply at first and then decreased gradually with an increase in shear rate. At low shear rate, SF and SS molecules start to move from rest, which increased the collision, intermolecular association and entanglement between macromolecules. With the shear rate continuously increased, the macromolecules began to move orderly along the circumferential shear direction, and the aggregation and entanglement between macromolecules decreased. Figure 3 shows the shear viscosity varied greatly among the groups. Whether shear viscosity was zero or maximum, pure SS was the highest. In our experiment, SS in the SF/SS_4_ group was almost completely removed, and its degumming rate compared with Na_2_CO_3_ degumming showed no difference. However, the shear viscosity of SF/SS_4_ was 14 times that of SF degummed by Na_2_CO_3_ (low molecular weight and large degree of dispersion). These results also showed that damage to the SF macromolecule chain was lower when degummed in boiling water than in Na_2_CO_3_. When the degumming time was reduced, the SS content and the hydrophilic groups in the SF/SS composites increased, leading to increased intermolecular association and more entanglement points between SF and SS macromolecules. Thus, the maximum value of viscosity was SS > SF/SS_1_ > SF/SS_3_ > SF/SS_4_ > SF. These results were consistent with those obtained by SDS-PAGE.

### 3.5. Molecular Conformation of SF/SS Composite Films 

The molecular conformation is the key factor in regulation of the mechanical properties and biological activity of a protein. Figure 4 shows the FTIR spectral curves of SF and SF/SS films. SF and SF/SS contain the same types of amino acids, and all the films showed characteristic bands at approximately 3277 cm^−1^, 1637 cm^−1^, 1513 cm^−1^, and 1233 cm^−1^ (Figure 4A), which were assigned to the N–H stretching vibration, C=O stretching vibration, N–H bending vibration, and C–N stretching vibration, respectively [44]. After ethanol treatment, the peak of amide I of all the SF/SS composite films shifted from 1637 cm^−1^ to 1618 cm^−1^, indicating the random coil changed to β-sheet conformation (Figure 4B). The molecular conformational content of all films was analyzed by Peakfit v4.1 software (SYSTAT, San Jose, CA, USA) and shown in Appendix A. The β-sheet content of the films was: SF/SS_4_ > SF/SS_3_ > SF > SF/SS_1_. This also indicated that boiling water degumming caused less damage to SF macromolecular chains, which contributed to the formation of the β-sheet. Therefore, the β-sheet content in the composite films following boiling water degumming was higher compared with the SF film degummed by Na_2_CO_3_. In the SF/SS_1_ film, more hydrogen bonds between SS and SF macromolecules hindered the formation of regular β-sheets between SF molecular chains, which resulted in decreased β-sheet content, especially in the ethanol treatment group.

### 3.6. Crystal Structure of SF/SS Composite Films

The H-chain of SF has 12 Gly-X repetitive structural domains and 11 short structural domains arranged alternately. The repetitive region contains many (AGSGAG)n with high copy number [45], which causes macromolecules to aggregate and form a dense crystalline structure. SS contains amorphous and α-helical structures with small crystalline regions [46]. Ethanol treatment can induce the transformation of crystalline structure of silk protein. As shown in Figure 5A, there were obvious silk I crystalline diffraction peaks at 12.1°, 19.9° (relatively sharper) and 24.7° for the SF film, and 12.1°, 19.9° (broad peak) for SF/SS composite films. After ethanol treatment, the crystalline structure of all films changed significantly (Figure 5B). A weak peak appeared at 9.1°, a sharp peak at 20.7°, and an obvious peak at 24.1°, which were silk II crystalline peaks. Nevertheless, the silk I crystalline diffraction peak at 12.1° did not disappear in the SF/SS composite films. There was no significant difference between the ethanol-treated films. 

### 3.7. Mechanical Properties of SF/SS Composite Films

The mechanical properties of a material are typically useful to further identify its applicability in specific tissue engineering. Scaffolds/films for bone tissue engineering must exhibit excellent mechanical properties, especially tensile or compressive strength compared to those used for soft tissue reconstruction. Therefore, we evaluated the strength and Young’s modulus of these SF/SS composite films in the wet state. Figure 6 shows the stress-strain curves of the SF/SS composite films. The breaking strength and breaking elongation of SF/SS composite films were significantly higher than that of pure SF film. The breaking strength of SF/SS_4_, SF/SS_3_ and SF/SS_1_ were 1.2, 1.6 and 1.4 times that of SF film, and the breaking elongation were 1.7, 2.3 and 2.4 times that of SF film, respectively (Appendix A). By contrast, SF/SS_3_ film showed the best strength and toughness. The content of SS had a significant effect on the tensile properties. The mechanical properties of silk protein-based materials or blends depend on their molecular weights, crystallinities, and compatibility between components. The higher the molecular weight, the mechanical properties of materials are stronger [47,48]. With decreased boiling water degumming time, the SS content increased which not only caused less damage to the SF macromolecular chains than Na_2_CO_3_ degumming, but also increased the bonding force between molecular chains and the relative slip of the SF molecular chain. The pure SF film showed highest Young’s modulus, lowest elongation, which resulted in a brittle material. These results indicated that the flexibility of SF/SS composite films was significantly better than SF films, which was due to the stronger hydrogen bonds and formed between SF and SS molecules.

### 3.8. Thermal Performance of SF/SS Composite Films

The thermal analysis reflects the thermal stability of material, which related to structural changes on polymers such as crystallization, and the higher the crystallinity, the higher the fastest thermal decomposition temperature [49]. As shown in Figure 7, two distinct regions of thermal weight loss were observed in the thermogravimetric curves of all samples. Slight mass loss was observed for all samples at 30–150 °C due to the evaporation of free and bound water. The greater the SS content, the higher the fastest thermal decomposition temperature. The order of the fastest thermal decomposition temperatures of the SF/SS composite films was SF/SS_1_ > SF/SS_3_ = SF/SS_4_ > SF, which were 311.2 °C, 310.8 °C, 310.8 °C and 308.3 °C, respectively, all of which were lower than that of SS at 327.4 °C. The results also showed that boiling water degumming (SF/SS_4_) reduced the damage to SF macromolecular chains and enhanced the intermolecular force, which together improved the mechanical properties, compared with Na_2_CO_3_ degumming. Although the thermal decomposition temperature of SS was higher than all other samples, the residual amount after decomposition was lower, which was probably related to the fact that the SS molecule is spherical, formed a strong intermolecular association and a weak intramolecular interaction force, while the SF molecule is fibrous and can self-assemble to form a dense β-sheet structure in the molecular chain. Moreover, the SF showed sharper decomposition peak, indicating more homogeneous thermal stability than SS [50], while SF/SS with less homogeneity components showed wide peak and shoulder.

### 3.9. Mineralization Properties of SF/SS Composite Films

#### 3.9.1. Surface Morphology and Ca-P Elements

During the mineralization process, Ca^2+^ from the SBF can bind to –COOH or –OH of SF/SS material via electrostatic interaction and coordination binding, and then Ca^2+^ which is enriched on the SF/SS material further attracts PO_4_^3−^ from the SBF to form nanocrystalline nuclei (Appendix A) [51]. As shown in Appendix A, nanocrystals began to appear on the film surfaces after mineralization for 1 day. At 3 days, it was evident that there were thin mineral layers on the film surfaces, and large spherical crystals appeared on the SF/SS composite films, but the crystals on the pure SF film surface were smaller and fewer. With increased mineralization time, the spherical crystals on all film surfaces grew further and aggregated into larger crystals, among which the number of crystals on the SF/SS_1_ film was largest, and the spherical crystals almost spread throughout the field of view after 7 d mineralization (Figure 8A). The spherical crystals on the surface of SF/SS_4_ were fewer than those on SF/SS_3_ and SF/SS_1_. Compared with the composite films, the nucleation rate on the surface of pure SF film was slower with the least number. These results indicated that increased SS content significantly promoted biomineralization.

As shown by EDX analysis, the distribution of Ca and P elements was observed on each mineralized film. After mineralization for 1 day, the SF/SS_1_ film showed more Ca elements (red, Appendix A) and P elements (green, Appendix A) on the surface than the other groups. After mineralization for 3 d, the distribution of Ca and P on all films increased, and completely covered the surface of SF/SS_1_. The number of bright spots was as follows: SF/SS_1_ > SF/SS_3_ > SF/SS_4_ > SF. After 7 d of mineralization, all films were completely covered by Ca and P elements uniformly (Figure 8B,C). In addition, obvious cluster distribution appeared on the SF/SS_1_ surface. The results suggested that the deposited mineral was a Ca-P salt, and that, the greater the SS content, the faster the nucleation and aggregation.

Figure 8D shows the atomic ratio of Ca and P element (Ca/P) on the films after 7 d of mineralization. The Ca/P values on the surface of composite films SF/SS_1_, SF/SS_3_, SF/SS_4_, and pure SF film were 1.82, 1.75, 1.67 and 1.76, respectively, which were similar to the Ca/P value of 1.67 for natural bone. The Ca/P value of commercial HAp nanoparticles was 2.05, which was higher than natural bone, and may be caused by the small amount of calcium oxide composition in the mineral phase [52]. This may also be the reason for the increased Ca/P values on SF/SS_1_ and SF/SS_3_ films with high SS content.

#### 3.9.2. Chemical Structure of Mineralized Products

Figure 9 shows the FTIR spectra of the mineralized products on SF/SS composite films. After mineralization for 1 d, with the exception of characteristic absorbance bands of amide I (1618~1670 cm^−1^ of C=O stretching vibration), amide II (1513~1550 cm^−1^ of N–H bending vibration) and amide III (1230~1245 cm^−1^ of C–N stretching vibration) derived from SF or SS (Figure 9A,C), new absorption peaks were observed around 1020 and 962 cm^−1^ assigned to the stretching vibrations of P–O, and around 599 and 558 cm^−1^ assigned to the bending vibrations of O–P–O, respectively [53], which were consistent with the characteristic peaks of HAp (Figure 9B). As the mineralization time increased, the characteristic peak of C–N stretching vibration disappeared, the N–H bending vibration and C=O stretching vibration then weakened successively, while the HAp-like characteristic peaks gradually increased (Figure 9D–F). Comparing the stretching vibration strength of P–O with C=O (amide I), the I_P–O_/I_C=O_ value of SF/SS composite films was greater than SF/SS_4_ and pure SF films, indicating that SS in SF/SS composite films contributed to the rapid mineralization of SF-based materials to form HAp crystal nuclei.

#### 3.9.3. Crystal Structure of Mineralized Products

Figure 10 shows the XRD patterns of all films after mineralization for different times and the crystallographic planes corresponding to the characteristic diffraction peaks of commercial HAp marked according to the standard card (JCPDS 09-0432). There were characteristic diffraction peaks of nanocrystalline HAp at 25.86°, 31.96°, 39.86°, 46.72° and 49.44°, corresponding to (002), (211), (310), (222) and (213) planes, respectively [54]. After mineralization for 1 d, all composite films still showed sharp diffraction peaks around 20.5°, similar to the non-mineralized composite films, which was attributed to the silk II crystalline structure. In addition, diffraction peaks appeared at 25.86° (002), 31.96° (211), and 39.86° (310), which were consistent with reference data corresponding to HAp. Moreover, the diffraction peak intensity of the (002) crystal plane of SF/SS_1_ was significantly stronger. With increased mineralization time, the intensity of the protein characteristic peak around 20.5° gradually decreased, while the corresponding characteristic diffraction peaks around 25.86° (002) and 31.96° (211) gradually increased, and the diffraction peaks of (222) and (213) planes appeared in sequence. Comparing the diffraction intensity of 31.96° with 20.5°, the I_31.96°_/I_20.5°_ value of SF/SS composite films was greater than pure SF films, indicating that SF/SS composites and pure SF material could induce the formation and growth of HAp-like crystal nuclei, and the presence of SS was conducive to the occurrence of this process. These results were consistent with those of the infrared spectrum.

### 3.10. Thermal Analysis of Mineralized Products

The mineralization ability and mineralized products were further analyzed by TG-DTG. Figure 11 shows the fastest thermal decomposition temperatures of SF/SS composite films after 7 d of mineralization to reflect the changes in thermal properties. All samples were observed to have a small mass loss at 80–100 °C, which was attributed to water evaporation. After mineralization for 7 d, the temperature corresponding to the fastest thermal decomposition rates of SF/SS_1_ and SF/SS_4_ were lower than pure SF film, which were 309.7 °C, 309.8 °C and 310.3 °C, respectively. It was speculated that the crystallinity of the films was destroyed because the -COOH of SS and SF was involved in mineralization, which accelerated the thermal decomposition of the films. Compared with the unmineralized films, the thermal decomposition mass loss of each film slowed down at 600 °C, due to the formation of HAp, which had a high thermal decomposition temperature. SF/SS_1_ (58.22%) had a higher residual mass than SF/SS_4_ (52.41%) and SF (53.90%) films, this was because the presence of SS accelerated mineralization on the films, and the greater the SS content, the more crystals were mineralized and deposited on the surface of the films.

## 4. Conclusions

In this study, we prepared a novel SF/SS composite film suitable for bone tissue engineering using a boiling water degumming process. This is a green and extremely simple preparation method. The molecular weight of silk protein degummed by this method was significantly larger than that by Na_2_CO_3_, indicating that boiling water degumming caused less damage to the SF macromolecular chain structure. In addition, the crystallinity of SF/SS composite films significantly increased with increased SS content, which had a positive impact on improving the mechanical properties of the films, in fact, the strength and toughness were significantly improved. At the same time, SF/SS composite films effectively induced mineral deposition, nucleation, growth and aggregation of HAp-like products, and the Ca/P values of the deposited minerals were similar to natural bone. The presence of SS helped to promote the mineralization of SF-based materials. Accordingly, the SF/SS composite film prepared in this study is expected to serve as a candidate material for bone tissue engineering.

## Figures and Tables

**Figure 1 polymers-14-02466-f001:**
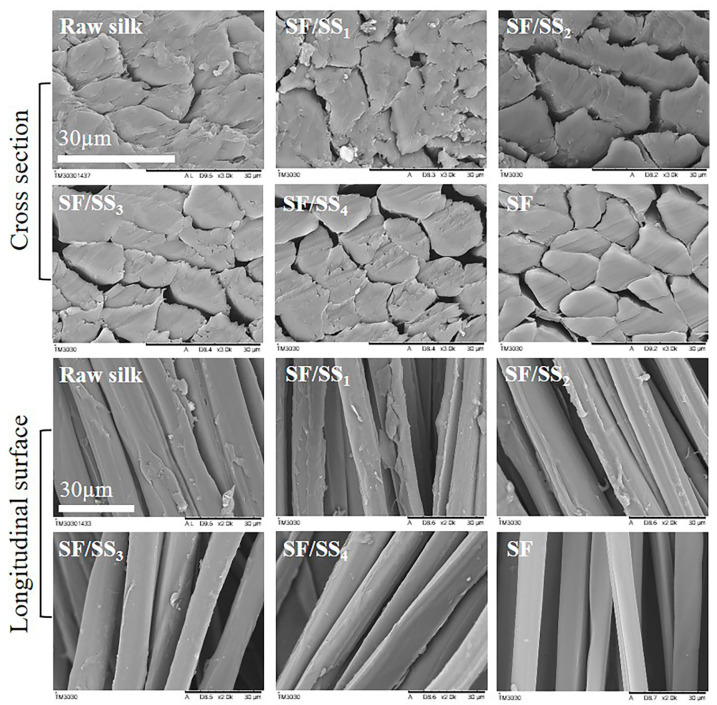
The microstructure of silk in different degumming degrees.

**Figure 2 polymers-14-02466-f002:**
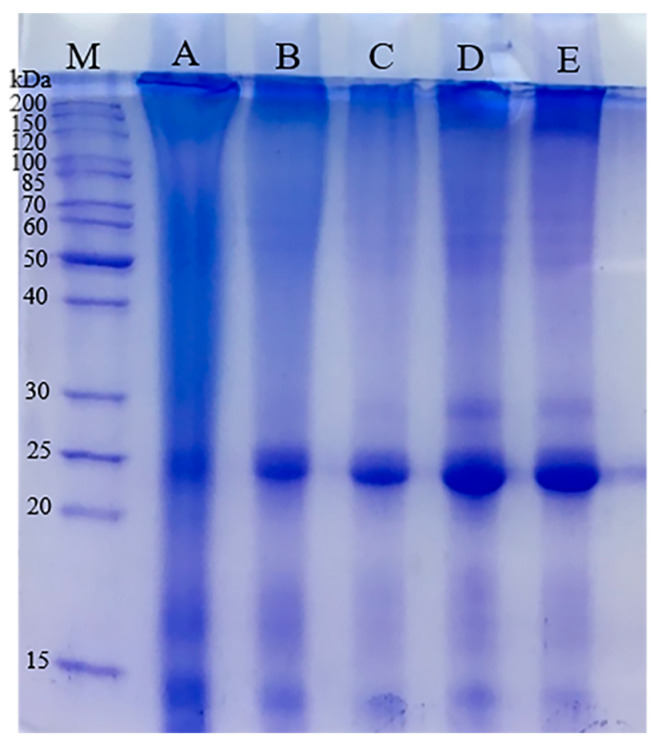
SDS-PAGE of SF/SS composite solutions. M: protein molecular weight standards; A: SF; B: SF/SS_4_; C: SF/SS_3_; D: SF/SS_2_; E: SF/SS_1_.

**Figure 3 polymers-14-02466-f003:**
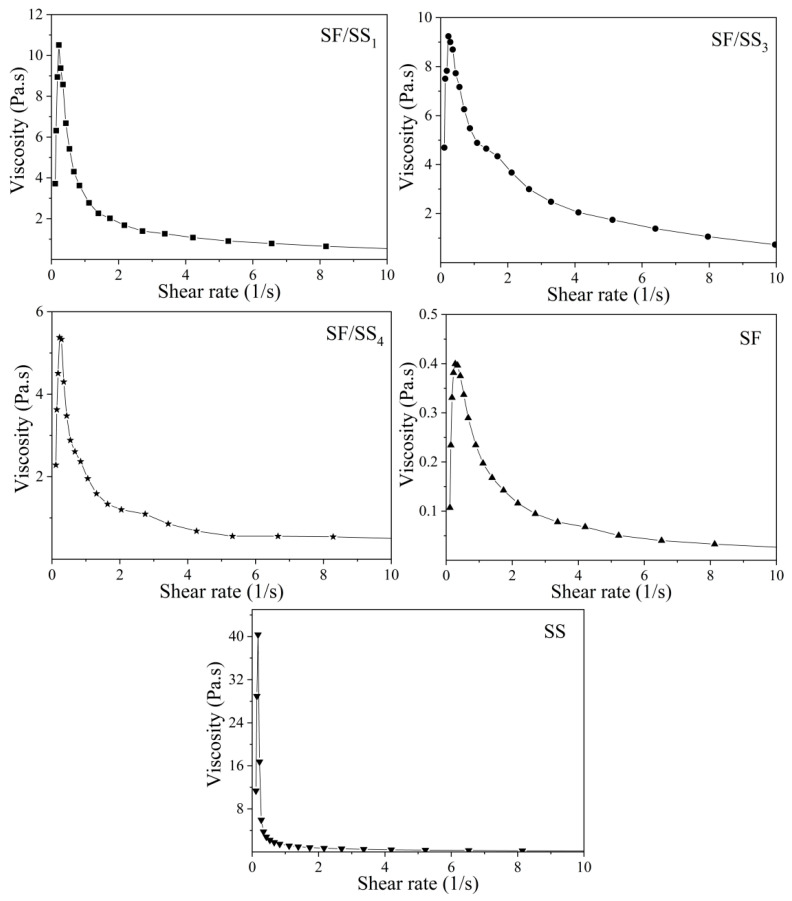
Rheological properties of SF/SS composite solutions and SS solution.

**Figure 4 polymers-14-02466-f004:**
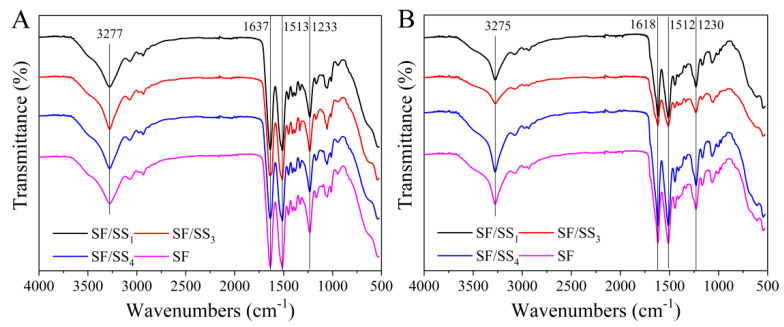
ATR−FTIR spectrum of SF/SS composite films. (**A**) No treatment, (**B**) Ethanol treatment.

**Figure 5 polymers-14-02466-f005:**
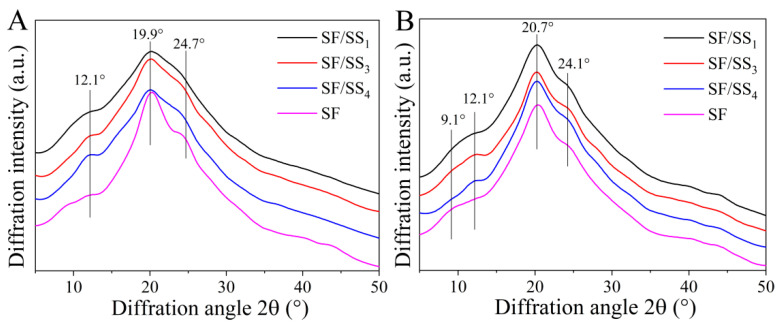
XRD patterns of SF/SS composites. (**A**) No treatment, (**B**) Ethanol treatment.

**Figure 6 polymers-14-02466-f006:**
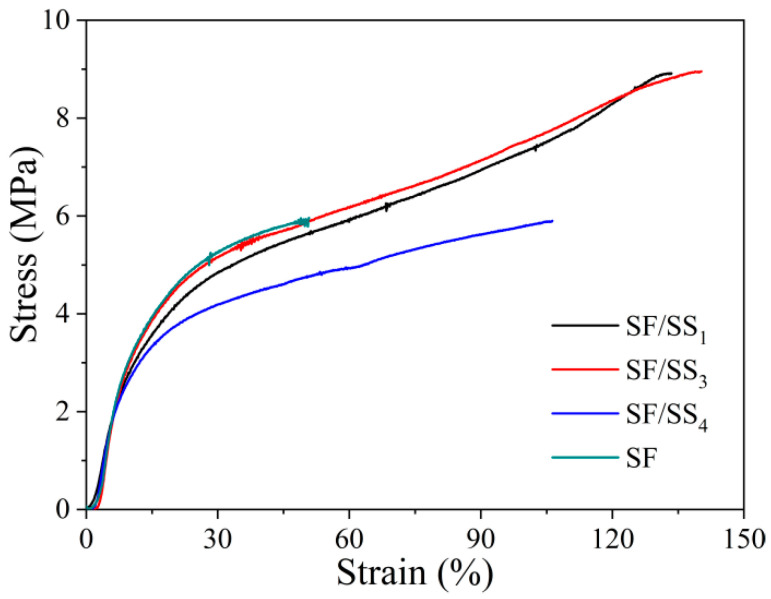
Stress-strain curves of SF/SS composite films.

**Figure 7 polymers-14-02466-f007:**
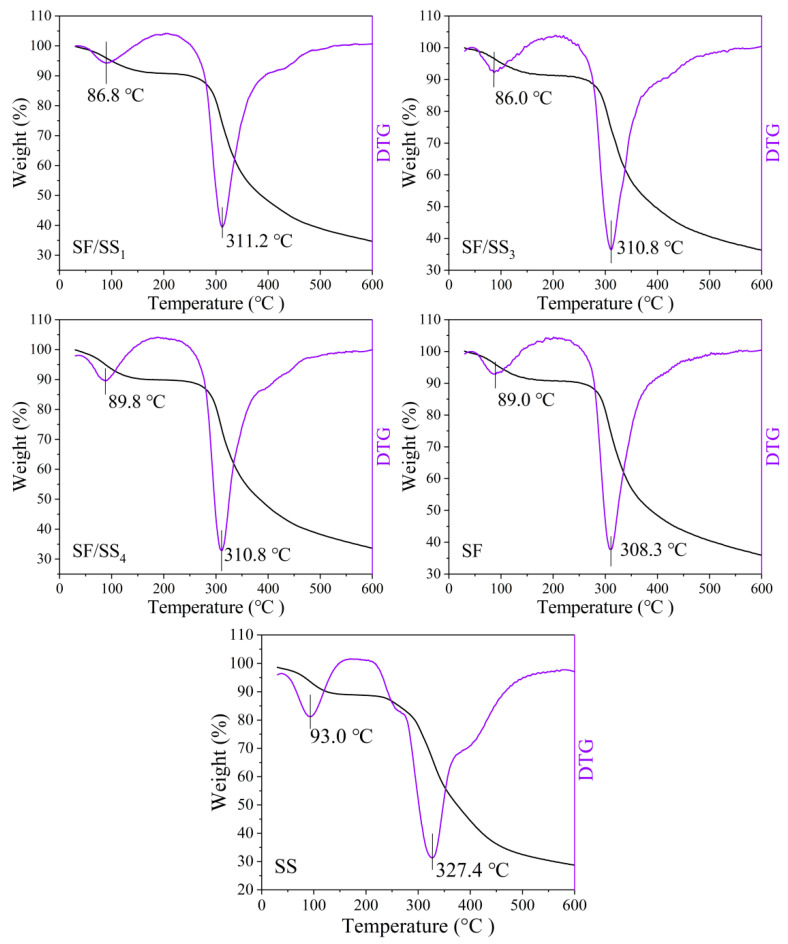
TG-DTG curve of SF/SS composite films.

**Figure 8 polymers-14-02466-f008:**
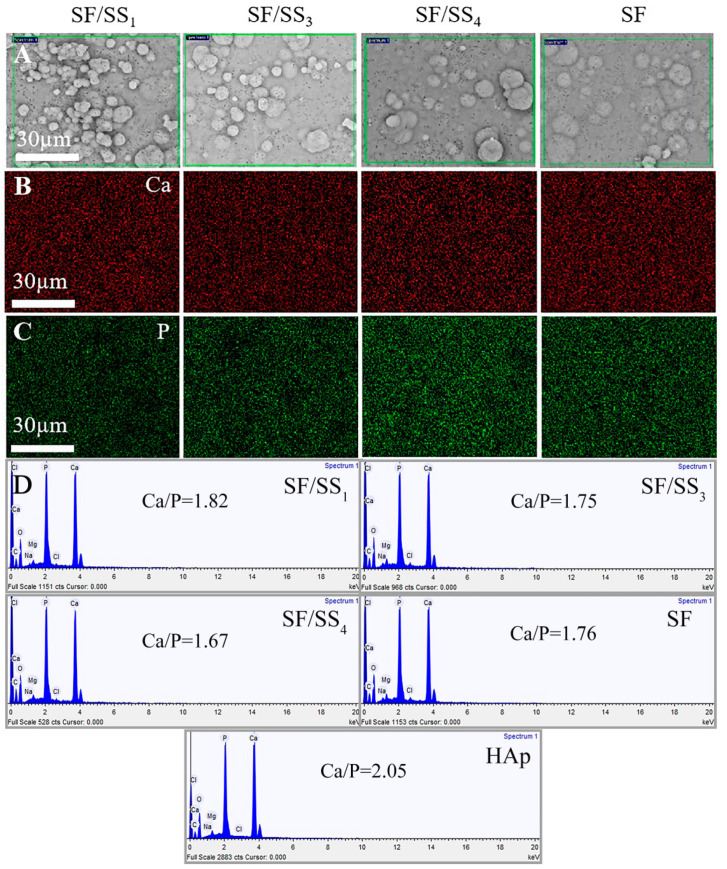
The distribution of growing crystals and elements after mineralization for 7 d. (**A**) SEM images of growing crystals, (**B**) Distribution of Calcium, (**C**) Distribution of Phosphorus, (**D**) Ca/P values.

**Figure 9 polymers-14-02466-f009:**
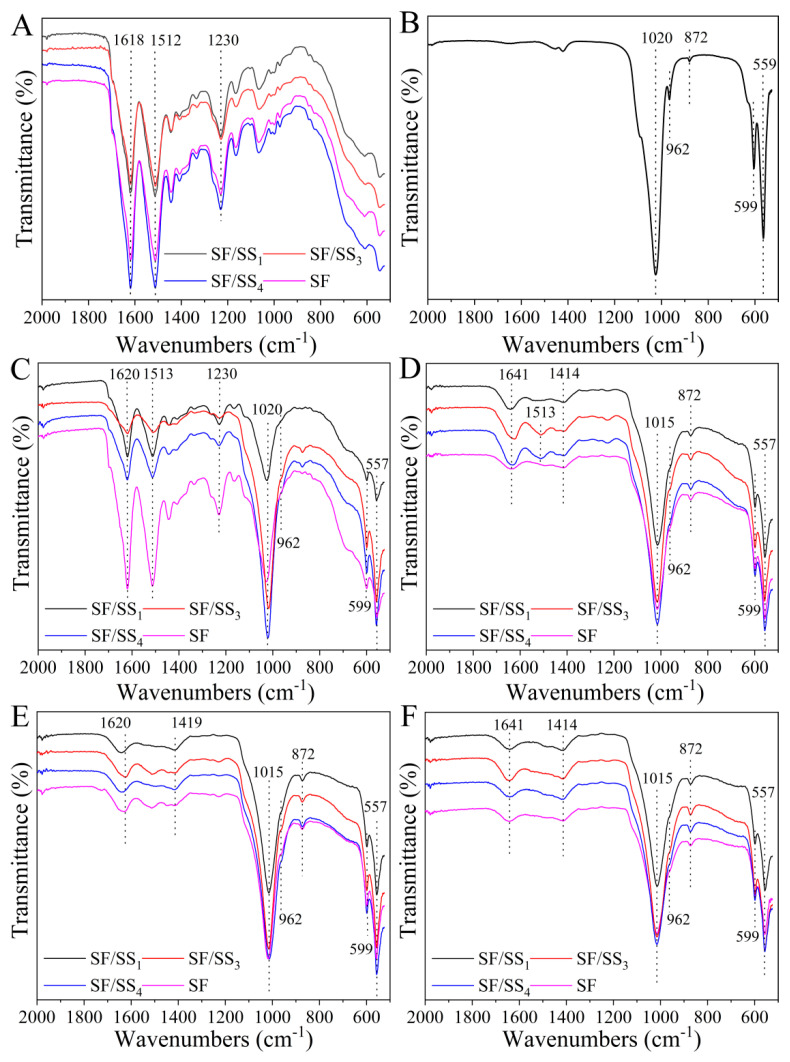
AIR-FTIR spectra of SF/SS composite films after mineralization. (**A**) Unmineralized, **(****B**) HAp, (**C**) 1 d, (**D**) 3 d, (**E**) 5 d, (**F**) 7 d.

**Figure 10 polymers-14-02466-f010:**
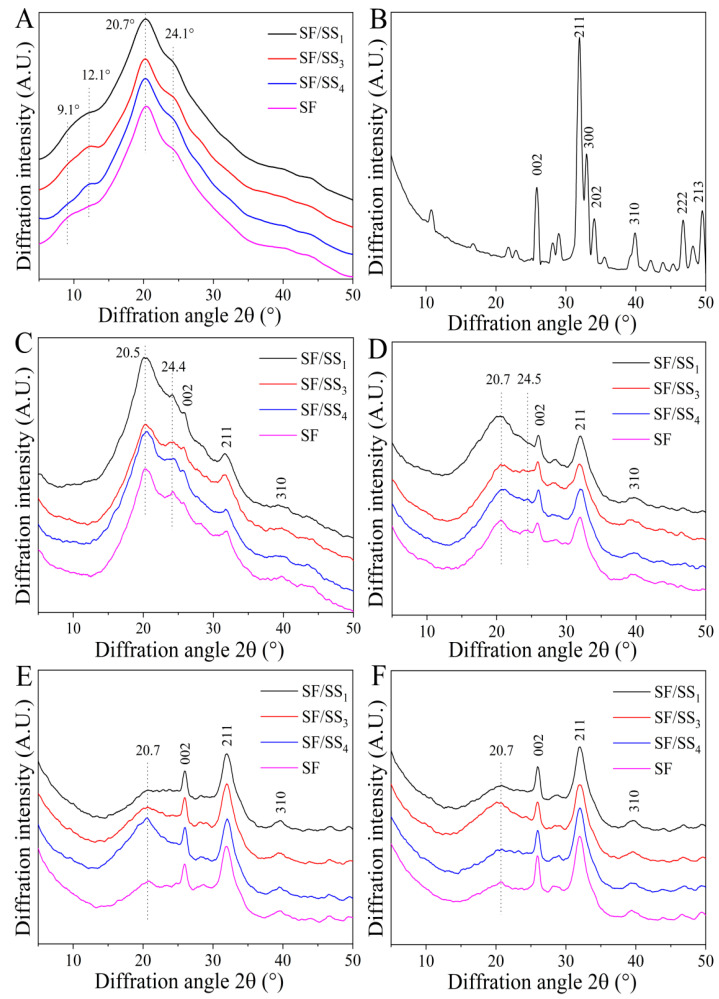
XRD patterns of the mineralized SF/SS composite films and pure HAp. (**A**) Unmineralized, (**B**) HAp, (**C**) 1 d, (**D**) 3 d, (**E**) 5 d, (**F**) 7 d.

**Figure 11 polymers-14-02466-f011:**
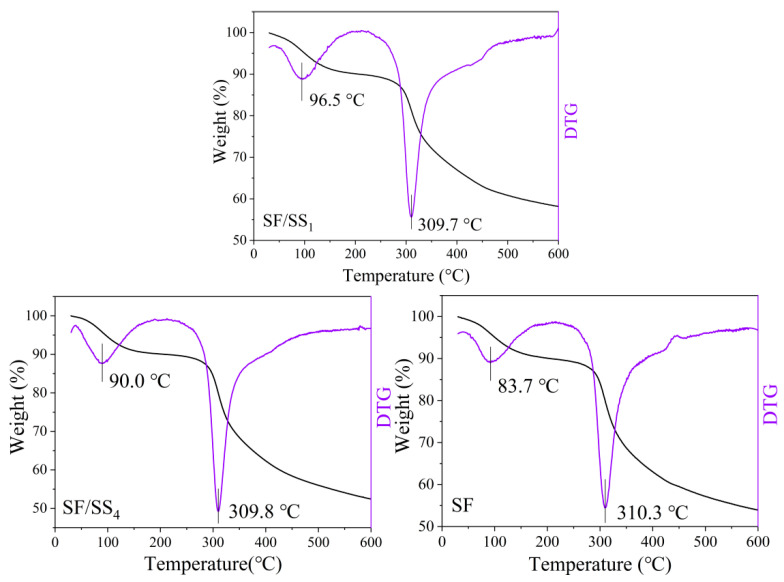
TG-DTG curve of SF/SS composite film after mineralization for 7 d.

**Table 1 polymers-14-02466-t001:** Degumming method and degumming rate of silk.

Degumming Method		Assignment	Degumming Rate (%)
Boiling water	1 h	SF/SS_1_	9
	2 h	SF/SS_2_	13
	3 h	SF/SS_3_	15
	5 h	SF/SS_4_	21
Na_2_CO_3_	1.5 h	SF	23

**Table 2 polymers-14-02466-t002:** Amino acid composition of SF/SS composites solutions (mol%).

Amino Acid	SF	SF/SS_4_	SF/SS_3_	SF/SS_2_	SF/SS_1_	SS
Asp	1.64	1.82	2.34	2.67	2.83	16.55
Thr	0.91	1.01	1.37	1.57	1.72	8.41
Ser	10.32	10.60	11.45	11.79	11.91	35.27
Glu	1.65	1.80	1.91	1.96	2.52	4.70
Gly	43.66	42.73	41.92	41.16	39.76	14.49
Cys	0.04	0.04	0.07	0.07	0.00	0.00
Ala	31.06	31.13	29.92	29.19	28.86	4.10
Val	2.23	2.50	2.32	2.38	2.62	3.24
Met	0.03	0.04	0.08	0.14	0.00	0.13
Ile	0.55	0.59	0.59	0.67	0.61	0.53
Leu	0.51	0.50	0.57	0.62	0.71	1.13
Tyr	5.10	5.06	4.96	4.98	5.15	2.85
Phe	0.77	0.78	0.80	0.75	1.01	0.53
Lys	0.25	0.34	0.39	0.43	0.40	2.65
His	0.23	0.21	0.29	0.29	0.30	1.39
Arg	0.45	0.50	0.61	0.69	0.71	2.98
Pro	0.55	0.59	0.38	0.60	0.91	1.06

## Data Availability

The data presented in this study are available on request from the corresponding author.

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
