# Peer review of "Enhanced Silk Fibroin/Sericin Composite Film: Preparation, Mechanical Properties and Mineralization Activity"

_polymers, 2022, doi:10.3390/polym14122466_

Round 1
Reviewer 1 Report
The article entitled "Enhanced silk fibroin/sericin composite film: Preparation, mechanical properties and mineralization activity" presents a new method for degumming and developing films from natural polymers, following a green route, which after minor adjustments must be accepted for publication, in this journal, Polymers.
The article must be placed in the journal's template and standards.
In item 2, a Materials sub-item must be included, indicating all reagents (and their data, purity, origin, etc.).
Replace the M and N units per mol/L throughout the text.
Insert in Table 1 the time for the process using Na2CO3.
Better detail the conditions used in the procedure of sub-item 2.3.
Check and standardize names and abbreviations of all amino acids mentioned in sub-item 3.2
The data presented in Table 2 needs to be discussed.
SF/SS2 system data was not presented in most results, why?
Figure 4 needs to be improved, and the most important regions highlighted.
In the discussion of mechanical properties, hydrogen bonds are indicated as the interaction between polymers, but in sub-item 3.9.1 it is mentioned that polymers can make electrostatic bonds with Ca2+. Other types of interactions between polymers cannot occur?
Authors should add illustrative schemes of material formation and mineralization.
Author Response
1. The article must be placed in the journal's template and standards.
Answer: This revision was based on the manuscript version revised by editorial office.
2. In item 2, a Materials sub-item must be included, indicating all reagents (and their data, purity, origin, etc.).
Answer: We have added a materials sub-item in the item 2.
3. Replace the M and N units per mol/L throughout the text.
Answer: We have replaced the M and N units per mol/L throughout the text.
4. Insert in Table 1 the time for the process using Na2CO3.
Answer: We have inserted the time for the process using Na2CO3 in the table 1.
5. Better detail the conditions used in the procedure of sub-item 2.3.
Answer: We have revised them.
6. Check and standardize names and abbreviations of all amino acids mentioned in sub-item 3.2.
Answer: Asp, Ser and Glu have been been explained in the introduction.
7. The data presented in Table 2 needs to be discussed.
Answer: We have added discussion to the data in the table 2.
8. SF/SS2 system data was not presented in most results, why?
Answer: We have explained this reason in the section 3.3 of the original manuscript. The reason is “As the molecular weight distribution of SF/SS1 and SF/SS2 were similar, we selected SF/SS1 for subsequent research”, so SF/SS2 system data was not presented in most results.
9. Figure 4 needs to be improved, and the most important regions highlighted.
Answer: We have enlarged the figure size and improved its resolution.
10. In the discussion of mechanical properties, hydrogen bonds are indicated as the interaction between polymers, but in sub-item 3.9.1 it is mentioned that polymers can make electrostatic bonds with Ca2+. Other types of interactions between polymers cannot occur?
Answer: The mechanical properties of the films are mainly determined by the chemical forces between the SF molecules. While sub-item 3.9.1 is mainly focused on the function of surface of polymer films, which is the chemical mechanism of induced mineralization, namely -COOH and -OH groups binding Ca2+ through electrostatic interaction and coordination binding. The mechanism of mineralization was explained in the introduction and supplemented in the sub-item 3.9.1.
11. Authors should add illustrative schemes of material formation and mineralization.
Answer: We have added illustrative schemes of material formation (Figure S1) and mineralization (Figure S2).
Reviewer 2 Report
This manuscript is well-written and provided a new approach to fabricate SF/SS composite film. The author provided thorough characterizations on fabricated SF/SS films and demonstrated that their potential usage as CaP mineralization scaffold. I will suggest accepting this manuscript after following concerns are addressed:
1. The image quality is low and makes it very hard to extract information from figures.
2. In Figure 8 SEM images, nanoparticles can be observed, and I would expect to see accumulated Ca and P signals accordingly in EDX. However, EDX images showed that Ca and P were almost uniformly distributed across the field. Could the author clarify this discrepancy?
3. In mineralization section, the author claimed, “The crystals on the surface of SF/SS4 were fewer than those on SF/SS3 and SF/SS1.” While it clear that SF/SS1>SF/SS4, the difference between SF/SS3 and SF/SS4 is not that obvious. I would recommend the author to conduct a static analysis on nanoparticles density on these images and provide a quantitative result to further support this statement.
Author Response
Reviewer #2
1. The image quality is low and makes it very hard to extract information from figures.
Answer: We have enlarged the size of each figure, and improved the resolution.
2. In Figure 8 SEM images, nanoparticles can be observed, and I would expect to see accumulated Ca and P signals accordingly in EDX. However, EDX images showed that Ca and P were almost uniformly distributed across the field. Could the author clarify this discrepancy?
Answer: The surface of the SF/SS films prepared in this article contained a large number of -COOH and -OH groups, which were uniformly distributed and fully covered. After mineralization, crystal nuclei will be uniformly distributed on the surface of the films, but there were still differences in nucleation and growth rate. In the original manuscript, we have indicated that “After 7 d (It was mistakenly written as “5 d” in the original manuscript) of mineralization, all films were completely covered by Ca and P elements uniformly (Figure 8B, C). In addition, obvious cluster distribution appeared on the SF/SS1 surface.”
3. In mineralization section, the author claimed, “The crystals on the surface of SF/SS4 were fewer than those on SF/SS3 and SF/SS1.” While it clear that SF/SS1>SF/SS4, the difference between SF/SS3 and SF/SS4 is not that obvious. I would recommend the author to conduct a static analysis on nanoparticles density on these images and provide a quantitative result to further support this statement.
Answer: Thanks very much for your considerate advice. This paper only preliminary studied the mineralization effect. We will further consider appropriate methods for quantitative analysis of crystal formation, growth and three-dimensional characteristics, and explore the regulation mechanism of mineralization.
Reviewer 3 Report
Section 1. - Provide clear objective in the last paragraph
Section 2.3 Add detail of solution components e.g., protein concentrations, additive, plasticizer.
Section 2.9 Add condition of test e.g., speed.
Add statistical analysis -> Section 2.12
Section 3.4 Explain reason/discuss about the first increasing followed by gradual decrease.
Fig. 4 - Difficult to see. Please enlarge the figure size and resolution.
Section 3.6 What does “improved by” mean? More degree of crystallinity?
“…crystalline structure of all film changed…” How?
Section 3.7 Add more discussion e.g.,
L9 Bumbudsanpharoke et al. (2022) indicated that mechanical strength of the polymeric blend films depended on several factors including microstructures, interaction and compatibility between polymers (doi.org/10.1016/j.foodchem.2021.131709) and crystallinity (doi.org/10.1016/j.foodcont.2021.108541).
L13 Moreover, degumming process removed foreign molecules which likely gave more homogeneity in the matrices which subsequently improved strength against applied external force (doi.org/10.1016/j.foodchem.2021.130956).
L16 Increasing degree of hydrolysis possibly produced higher amounts of small molecular weight protein which plasticized matrices and improved elongation (doi.org/10.1016/j.fpsl.2021.100787).
Section 3.8 Recheck the meaning of the first sentence.
Add more discussion e.g., Moreover, the SF showed sharper degradation peak, indicating more homogeneous thermal stability than SS (doi.org/10.1016/j.fpsl.2022.100844), while SF/SS with less homogeneity components showed wide peak and shoulder.
Fig. 8 Revise for more contrast. Fig. D - cannot see the number. The size should be enlarged.
Author Response
Reviewer #3
1. Section 1. - Provide clear objective in the last paragraph.
Answer: We have revised.
2. Section 2.3 Add detail of solution components e.g., protein concentrations, additive, plasticizer.
Answer: We have added concentrations and volume of solution components and other parameters.
3. Section 2.9 Add condition of test e.g., speed.
Answer: We have added test condition.
4. Add statistical analysis -> Section 2.12.
Answer: We have added it.
5. Section 3.4 Explain reason/discuss about the first increasing followed by gradual decrease.
Answer: We have explained the reason in the section 3.4.
6. Fig. 4 - Difficult to see. Please enlarge the figure size and resolution.
Answer: We have enlarged the figure size and improved its resolution.
7. Section 3.6 What does “improved by” mean? More degree of crystallinity?
Answer: We want to express that ethanol treatment can induce the transformation of crystalline structure of silk protein, and have revised the express in the section 3.6.
8. “...crystalline structure of all film changed...” How?
Answer: Yes, the crystalline structure of all films changed from silk I to Silk II after ethanol treatment.
9. Section 3.7 Add more discussion e.g.
L9 Bumbudsanpharoke et al. (2022) indicated that mechanical strength of the polymeric blend films depended on several factors including microstructures, interaction and compatibility between polymers (doi.org/10.1016/j.foodchem.2021.131709) and crystallinity (doi.org/10.1016/j.foodcont.2021.108541).
L13 Moreover, degumming process removed foreign molecules which likely gave more homogeneity in the matrices which subsequently improved strength against applied external force (doi.org/10.1016/j.foodchem.2021.130956).
L16 Increasing degree of hydrolysis possibly produced higher amounts of small molecular weight protein which plasticized matrices and improved elongation (doi.org/10.1016/j.fpsl.2021.100787).
Answer: Thank you very much for recommending literatures to us. We have added the discussion and relevant literatures in the section 3.7.
10. Section 3.8 Recheck the meaning of the first sentence.
Add more discussion e.g., Moreover, the SF showed sharper degradation peak, indicating more homogeneous thermal stability than SS (doi.org/10.1016/j.fpsl.2022.100844), while SF/SS with less homogeneity components showed wide peak and shoulder.
Answer: We have revised the first sentence and added the discussion in the section 3.8.
11. Fig. 8 Revise for more contrast. Fig. D - cannot see the number. The size should be enlarged.
Answer: We have enlarged the size of Fig. 8 and improved its contrast.
Round 2
Reviewer 3 Report
The manuscript has been improved.